# Coupled C and N turnover in a dynamic pore scale model reveal the impact of exudate quality on microbial necromass formation

Maximilian Rötzer[1,4], Henri Braunmiller[2], Eva Lehndorff[3], Nadja Ray[1,4], Andrea Scheibe[3], and Alexander Prechtel[1]

[1]Modeling and Numerics, Department of Mathematics, University of Erlangen–Nürnberg, Erlangen, Germany
[2]Root-Soil Interaction, School of Life Sciences, Technical University of Munich, Freising, Germany
[3]Soil Ecology, Bayreuth Center of Ecology and Environmental Research (BayCEER), University of Bayreuth, Bayreuth, Germany
[4]Mathematical Institute for Machine Learning and Data Science, Catholic University of Eichstätt-Ingolstadt, Ingolstadt, Germany

**Correspondence:** Maximilian Rötzer (maximilian.roetzer@fau.de)

**Abstract.** The adequate quantification of soil organic carbon (SOC) turnover is a pressing need for improving soil health and advancing our understanding of climate dynamics. It is controlled by the complex interplay of microbial activity, availability of carbon (C) and nitrogen (N) sources, and the dynamic restructuring of the soil's architecture. Accurate modeling of SOC dynamics therefore requires the representation of these processes at small spatial scales. We present a mechanistic, spatially explicit pore-scale model, which couples enzymatic degradation of particulate organic matter (POM), microbial necromass and root exudates with microbial growth and turnover, C respiration and N cycling depending on the C/N ratios of the different organic carbon sources. It is combined with a cellular automaton model for simulating soil structure dynamics including the stabilization of soil particles, POM or microbial necromass via organo-mineral associations. The virtual soil simulations use parameters from rhizosphere experiments – without parameter fitting – to explore the influence of (i) soil structural heterogeneity and connectivity, (ii) N limitation, and (iii) necromass formation on SOC storage. Our results demonstrate that evolving soil architecture and pore connectivity control substrate accessibility, creating micro-scale hot and cold spots for microbes. N availability consistently co-limits microbial growth, while a favorable C/N ratio of root exudates substantially reduces respiration and increases CUE over extended periods. Necromass emerges as a persistent SOC pool, as N derived from from short-term root exudation pulses promotes biomass growth and is subsequently converted into slowly degradable necromass, which can be physically protected through occlusion. These findings are consistent with lab experiments and additionally provide insights into the spatial and temporal dynamics of the drivers of carbon turnover.

## 1 Introduction

Soil organic carbon (SOC) turnover is a fundamental component of the global carbon cycle with far-reaching implications for soil health and climate dynamics. In their review Vogel et al. (2024) emphasized the need for an explicit representation of biological processes in models to adequately capture soil functions such as the turnover of organic matter, N cycling, or soil structure formation, and state that this is a prerequisite to arrive at the urgently needed predictive models understanding

perturbations caused by soil management or current changes in climatic conditions. Varney et al. (2022) compared soil carbon simulations in current earth system models and concluded that much of the uncertainty can be attributed to the simulation of below-ground processes, and greater emphasis is required on improving their representation in future developments of models.

Accurate modeling of SOC dynamics requires a nuanced understanding of a range of interconnected biological and physical processes occurring at small spatial scales (He et al., 2024; Pot et al., 2021).

We aim to address several challenges: First, the highly heterogeneous soil structure and its dynamically evolving connectivity intricately link microbial activities and stabilization mechanisms. As noted, e.g. by Sokol et al. (2022), a mechanistic understanding of the processes governing the formation and decomposition of mineral-associated organic matter (MAOM)

is critical. At the small scale, aggregates can be conceptualized as microbial reactors whose evolving properties need to be explicitly represented in bottom-up modeling (Dungait et al., 2012; Wang et al., 2019). The microbial turnover of organic matter (OM), the subsequent formation of gluing spots and the build-up of organo-mineral associations are central to soil structure formation, and may on the other hand lead to physical protection of particulate organic matter (POM) (Totsche et al., 2018). Thus, structural dynamics and biological processes cannot be treated independently (Philippot et al., 2024). Second, the

soil-microbe system exhibits distinct responses to different slow or fast-cycling carbon/nitrogen sources. Besides plant-derived POM, microbial necromass can contribute a substantial fraction of SOC and must be explicitly taken into account (Chandel et al., 2023; Liang et al., 2019; Kästner et al., 2021). In the rhizosphere root exudates constitute an important and easily accessible carbon source which creates 'hot moments' and 'hot spots' for microbes (Baumert et al., 2018; Wiesenbauer et al., 2023). However, largely due to limitations of experimental techniques, quantifying their impact on aggregation and carbon turnover

in soils remains challenging (Gregory et al., 2022). Third, microbial growth is governed by the availability of resources and the specific microbial C/N demand, while microbial decomposition again feeds back on the overall C and N distribution in soils (Zechmeister-Boltenstern et al., 2015; Simon et al., 2024). This interplay should be incorporated in process-based models to improve our understanding of the dynamic nutrient cycling and microbial C/N turnover in soils (Kaiser et al., 2014; Zechmeister-Boltenstern et al., 2015; Sokol et al., 2022). He et al. (2024) emphasized that the link between microbial car-

bon use efficiency (CUE) (see (11) below) and SOC is mediated by processes that stabilize microbial necromass within soil aggregates or promote its association with mineral surfaces.

The complexity of soil systems necessitates mathematical approaches to study the interplay of the underlying mechanisms and enable more precise predictions of CUE responses under diverse environmental scenarios. As pointed out in several reviews (Baveye et al., 2018; Pot et al., 2021; Vogel et al., 2022), only explicit spatial and image-based modeling at the pore scale, and

50 not pool or lumped parameter models are capable to capture the interaction among soil architecture, microbial activity and OM turnover. Note that within such an approach, it is not necessary to a priori partition POM (or necromass) into fixed, less or more stable fractions or pools as, e.g. MAOM, instead the degradation rate of each POM particle is modified according to its degree of occlusion. This variation is inherent in the dynamic, spatially explicit model concept. The variability in soil structure as a consequence of the self organization of soil aggregates was first modeled by Crawford et al. (1997) and pursued further

simulating aggregate formation as a function of binding surfaces (of chemical or microbial origin) in Crawford et al. (2012). Subsequently, Ray et al. (2017) and Rupp et al. (2018b) developed a comprehensive cellular automaton method (CAM) to

simulate aggregate formation in a spatially and temporally explicit way. Zech et al. (2022b) applied a CAM to quantify the simultaneous structure formation/disaggregation and OM turnover and study the drivers texture, decomposition rate, and POM input. They revealed that the addition of fresh organic matter to soil may induce a short-term primed decomposition of native OM due to the creation of new gluing spots leading to the simultaneous break-up of older organo-mineral associations and the subsequent decomposition of POM. The first modeling study integrating root growth and exudation, microbial growth, particle surface alterations, aggregate dynamics and C turnover on the pore scale within a unified CAM framework was presented in Rötzer et al. (2023). Their findings indicate that exudates, although degraded within a few days, are important drivers for structure formation, leading to a more persistent stabilization of aggregate structures via an increased amount of gluing spots even after root degradation.

In the present work, we extend the pore scale model for coupled carbon turnover, microbial dynamics and structure evolution (Zech et al., 2022b; Rötzer et al., 2023). We account for different carbon sources (POM, necromass, and exudates) and include the interaction between C and N. We employ a biogeochemical model following (Manzoni and Porporato, 2009; Manzoni et al., 2012; Kaiser et al., 2014) and consider the specific demand for C and N of the microbial biomass. It takes into account that the microbial turnover creates organo-mineral interfaces with enhanced surface reactivity, the so-called gluing spots which enhance the formation of aggregates (Bucka et al., 2019). This surface conditioning effect of organic compounds on mineral surfaces is temporally limited, but may be locally retained even after degradation of the OM coating (see Zech et al. (2022b) for more details and references). The microbial turnover of the slowly decaying POM and microbial necromass and easily available carbon sources as exudates are studied in scenarios assuming different C/N ratios and concentrations. We elucidate the impact of changing soil structures and of N dynamics on microbial C turnover. We further evaluate the importance of necromass for the stabilization of carbon. We quantify our results e.g. in terms of CUE as our simulations are able to provide quantitative data in contrast to experiments.

## 2 Materials and methods

We introduce a process-based, spatially and temporally explicit micro-scale model that describes C and N pools and fluxes in soils along with the decomposition of particulate organic matter (POM), the development of microbial bio- and necromass, and simultaneous aggregate turnover. The latter is a consequence of the surface properties of soil particles, which are altered by the creation of gluing spots by microbes and organic remnants. The dynamic re-arrangement of primary soil particles and soil micro-aggregates impacts the spatial accessibility of OM and consequently its degradation rates. We also account for the input of root exudates as an easily available carbon source. These different model components are depicted in the conceptual drawing in Fig. 1, their interdependence and the transformation between them in Fig. 2. The relations are specified in more detail in Section 2.2.

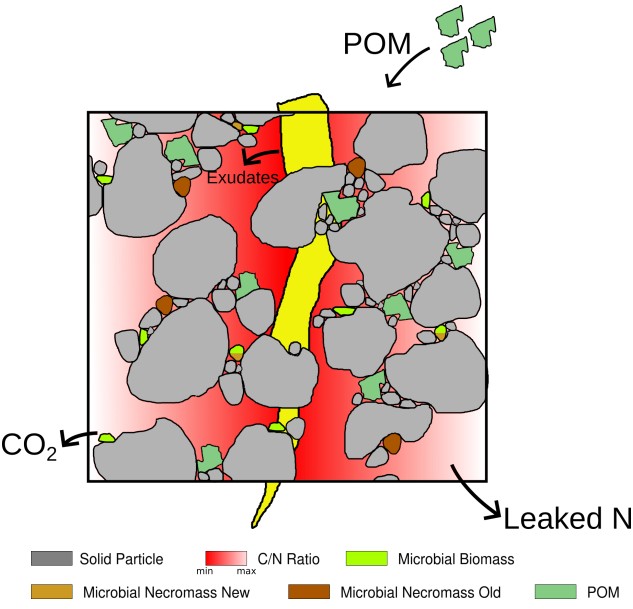

**Figure 1.** Illustration of the components of the model. The drawing includes a root (yellow) which is not taken into account explicitly as a model component, only its exudates.

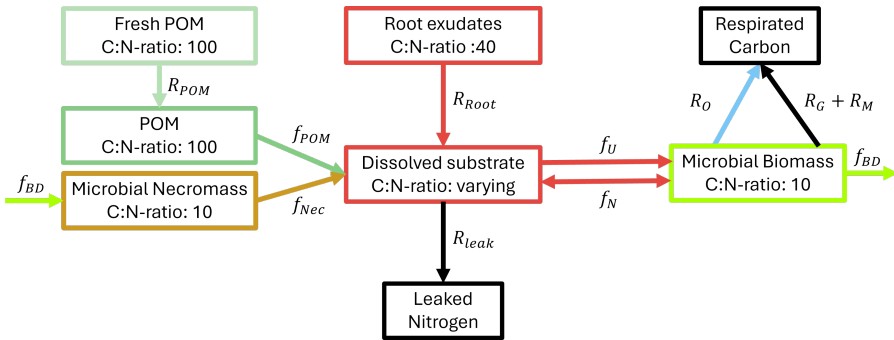

**Figure 2.** Conceptual model of the microbial dynamics with distinct carbon and nitrogen pools and fluxes between them.

## 2.1 Cellular automaton and soil restructuring

To account for the spatial distribution and re-arrangement of model components, we conduct our investigations in the flexible framework of a cellular automaton method extending the work of Zech et al. (2022b) and Rötzer et al. (2023). Within the CAM, the single-layered 2.5D computational domain is discretized into voxels (cells) of $2\,\mu m \times 2\,\mu m \times 1\mu m$, the smallest entities of the model. They have different (possibly co-existing) states such as solid, pore, POM, biomass or necromass, see Fig. 3. The solid phase consists of water-stable, inseparable, but mobile primary soil particles and micro-aggregates, see Appendix Section A1 for a detailed description. The pore space is assumed to be either saturated with soil solution carrying the dissolved

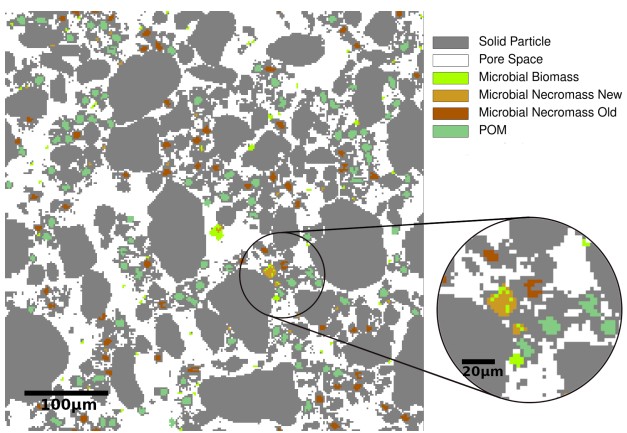

**Figure 3.** Sketch of the 2D discretized computational domain of 250x250 pixels with a resolution of 2 μm and different cell types. Edge types are not displayed for the ease of presentation.

substrates carbon and nitrogen, or occupied by solid microbial necromass or POM. As such oxygen limitations are disregarded due to small spatial scales. Note that different saturations and oxygen concentrations were investigated by Zech et al. (2024). These states may also co-exist. A key component of the model is the (dis-)aggregation of mobile particles according to different surface attractions. This concept is elaborated in detail in Rötzer et al. (2023) and Zech et al. (2022b), see also the Appendix. This attractiveness includes that with increasing size, a decreasing fraction of the soil particles' surfaces is marked as reactive. At these reactive surfaces, POM particles may attach preferably undergoing microbial degradation (Witzgall et al., 2021). The concept of reactive surfaces thus includes gluing spots originating from microbial remnants and occurring on temporary OM covered mineral surfaces. These gluing spots enhance the formation of aggregates (Bucka et al., 2019). As this surface conditioning of mineral surfaces by organic compounds may be locally retained even after degradation of the OM coating (Kleber et al., 2007), the term memory edges for mineral surfaces after degradation of OM has been introduced in Zech et al. (2022b). By interacting with the mineral surfaces these gluing spots strengthen the adhesive bonds between soil particles. Following Nazari et al. (2022), extracellular polymeric substances (EPS) and root exudates, such as mucilage, follow the same principle in enhancing the adhesion of mineral surfaces in soil. As they are both gelatinous high-molecular-weight substances, both substances create equally favorable hot-spots for the development of gluing surfaces.

### 2.2 C and N turnover

We extend the models in Zech et al. (2022b) and Rötzer et al. (2023) to incorporate N fluxes, take into account microbial growth dependent on C and N availability, and include necromass evolution as an important OM pool. Following the review by Manzoni and Porporato (2009) and Kaiser et al. (2014) on C/N pathways in soils, we consider the following model components directly related to different C and N pools with specific C/N ratios:

- Particulate organic matter (POM), rich in C with C/N=100, which is enzymatically degraded to dissolved C and N by microbial biomass

- Microbial biomass with C/N=10 is build up consuming dissolved C and N and is degraded to microbial necromass

- Microbial necromass with C/N=10, which is enzymatically degraded to dissolved C and N by microbial biomass

- Root exudates with C/N=40 directly contribute to dissolved C and N

- Soil solution with dissolved organic C and (in-)organic N (dynamic C/N ratio) is taken up by microbial biomass for growth and maintenance

- C and N outflows due to respiration and leakage, in case that the demand of C and N by microbial biomass is not balanced

We follow a concentration-based approach and, opposed to Manzoni and Porporato (2009), where the soil solution consists of distinguishable organic and inorganic N, combine the N assimilation pathways, the direct organic uptake and mineralization–immobilization. We assume that the pools microbial biomass, necromass, and POM are strictly homeostatic, i.e. they keep their C/N ratios constant over time, and constrain the C/N ratios of microbial biomass and necromass to be equal. Contrarily, dissolved C and N can be assimilated and diffuse independently from each other, i.e. the C/N ratio there is variable. We now state the underlying model equations - a system of differential equations - describing the evolution of the different C and N concentrations in the respective C and N pools and the fluxes between these pools as depicted in Fig. 2. The unknowns and reaction rates are summarized in Table 1 and 2 and the related parameters are specified in Table 4.

**Table 1.** State Variables (unknowns) of the model.

| Variable | Description |
|----------|-------------|
| $C_B$ | C in microbial biomass |
| $C_{Nec}$ | C in microbial necromass |
| $C_{CO_2}$ | Accumulated C from $CO_2$ respired by the microbes |
| $C_{POM}$ | C in particulate organic matter |
| $C_S$ | Organic C substrate dissolved in soil solution (DOC) |
| $N_B$ | N in microbial biomass |
| $N_{leak}$ | Accumulated leaked N, no re-entry into the system |
| $N_{Nec}$ | N in microbial necromass |
| $N_{POM}$ | N in particulate organic matter |
| $N_S$ | N dissolved in soil solution |

**Table 2.** Reaction rates of the model and their definition. The fluxes $f$ quantify the transformation of C and N from one pool to another while the rates denoted by $R$ represent a (external) loss or source, accounting for processes such as respiration, leakage, fresh POM input or root exudation.

| Rate | Description |
|------|-------------|
| $f_U = \nu_{C_S} \frac{C_S}{C_S + k_{C_S}} C_B$ | Microbial uptake of C |
| $f_N = \frac{f_U - R}{(C/N)_B}$ | Microbial assimilation or mineralization of N |
| $f_{POM} = k_{POM} \left(0.5 + \frac{C_B^{total}}{C_B^{init}}\right) \frac{s_f^p}{s^p} C_{POM}$ | Decomposition of POM |
| $f_{Nec} = k_{Nec} \left(0.5 + \frac{C_B^{total}}{C_B^{init}}\right) \frac{s_f^p}{s^p} C_{Nec}$ | Decomposition of necromass |
| $f_{BD} = k_{BD} C_B$ | Microbial decay |
| $R_O = \partial_t (C_B - \left(\frac{C}{N}\right)_B N_S)$ or 0 | Overflow respiration, see Appendix Eq. (A2) |
| $R_G = k_G f_U$ | Growth respiration |
| $R_M = k_M C_B$ | Maintenance respiration |
| $R = R_G + R_M + R_O$ | Total respiration |
| $R_{Leak} = k_{leak} N_S$ | Leaked N |
| $R_{POM}$ | Discrete input in time: addition of fresh POM particles |
| $R_{Root}$ | Input function of time for root exudation |

$$\frac{\partial C_B}{\partial t} = f_U - R_G - R_M - f_{BD} - R_O \tag{1}$$

$$\frac{\partial N_B}{\partial t} = \frac{\partial C_B}{\partial t} \cdot \frac{1}{(C/N)_B} \tag{2}$$

$$\frac{\partial C_S}{\partial t} - \nabla \cdot (D_S \nabla C_S) = f_{POM} + f_{Nec} - f_U + R_{Root} \tag{3}$$

$$\frac{\partial N_S}{\partial t} - \nabla \cdot (D_S \nabla N_S) = \frac{f_{POM}}{(C/N)_{POM}} + \frac{f_{Nec}}{(C/N)_{Nec}} - f_N + \frac{R_{\text{root}}}{(C/N)_{\text{root}}} - R_{\text{leak}} \tag{4}$$

$$\frac{\partial C_{Nec}}{\partial t} = f_{BD} - f_{Nec} \tag{5}$$

$$\frac{\partial N_{Nec}}{\partial t} = \frac{f_{BD}}{(C/N)_B} - \frac{f_{Nec}}{(C/N)_{Nec}} \tag{6}$$

$$\frac{\partial C_{POM}}{\partial t} = -f_{POM} + R_{POM} \tag{7}$$

$$\frac{\partial N_{POM}}{\partial t} = \frac{-f_{POM}}{(C/N)_{POM}} + \frac{R_{POM}}{(C/N)_{POM}} \tag{8}$$

$$\frac{\partial C_{\text{CO}_2}}{\partial t} = R_G + R_M + R_O \tag{9}$$

$$\frac{\partial N_{\text{leak}}}{\partial t} = R_{\text{leak}} \tag{10}$$

The reaction and transport system Eq. (1) – Eq. (10) determines how C and N cycles between microbial biomass ($C_B$ and $N_B$), microbial necromass ($C_{Nec}$ and $N_{Nec}$), dissolved substrates in the pore space ($C_S$ and $N_S$) and organic soil compounds such as particulate organic matter ($C_{POM}$ and $N_{POM}$). We also track the amount of C that is transformed to CO$_2$ as $C_{\text{CO}_2}$ and the leaked N as $N_{leak}$. In the review by Manzoni and Porporato (2009), in which a theoretical framework of well-established microbial biomass balance equations for the soil substrate–decomposer dynamics considering both, the microbial stoichiometry and the impact of limiting nutrients, is introduced. By following this concept the evolution of C in biomass, Eq. (1), depends on the C uptake from soil solution ($f_U$), the growth ($R_G$) and maintenance ($R_M$) respiration, the decay to necromass ($f_{BD}$), and a potential overflow mechanism ($R_O$). The flux $f_U$ describes the amount of C taken up from soil solution by the microbial biomass, modeled via a Monod kinetics. The growth respiration rate $R_G$ is assumed to be proportional to $f_U$. The basal metabolic turnover of microbes is summarized as maintenance respiration $R_M$ and is modeled as first-order kinetics. The decay term $f_{BD}$ accounts for the limited lifetime of microbial biomass and its transformation into microbial necromass thereafter. The overflow rate $R_O$ finally reflects the release of excess C from the system in the case of N limitation, see Appendix, Section A4 for a more detailed description. Diffusion–reaction equations for $C_S$, Eq. (3), and $N_S$, Eq. (4), describe the uptake from, release into and diffusive transport of C and N within the soil solution. We account for sources, specifically root exudation $R_{\text{Root}}$, hydrolysis of POM $f_{POM}$ and necromass $f_{Nec}$ and sinks via microbial uptake $f_U$ and leaked dissolved N $R_{\text{leak}}$. In Eq. (4), the term $f_N$ can be positive or negative, representing the assimilation or mineralization of $N_S$ as necessary to balance the C demand of the microbial biomass respecting its constant C/N ratio, see discussion above. In detail, assuming that exoenzymes released by the microbes are spatially homogeneous distributed in the soil, the enzymatic hydrolysis of POM $f_{POM}$ and necromass $f_{Nec}$

**Table 3.** Particle size distribution of the SPE loam.

| Feret diameter | 1-2 µm | 2-6 µm | 6 -20 µm | 20 - 63 µm | 63 - 200 µm |
|---|---|---|---|---|---|
| Contribution | 19.5 % | 5.2 % | 16.4 % | 26.3 % | 32.6 % |

follow first-order kinetics. Here, the turnover coefficients were adapted to represent two decomposition processes suggested by Chandel et al. (2023): Firstly, the rate coefficients are a function of microbial biomass. Assuming a basal turnover rate $k_{POM}$ and $k_{Nec}$, which include physical and chemical processes, the decay rate depends linearly on the total amount of microbial biomass in the domain $C_B^{total}$, normed by a factor $C_B^{init}$, the initial total amount of microbial biomass. As we mainly aim to differentiate between the impact of the easily available carbon source due to root exudates and the slowly cycling POM and necromass, we assume the same basal turnover rate for both POM and necromass. Secondly, we take into account that the degradation requires access of the POM particles to the pore space, and that the lack of this access decreases the rate (physical occlusion, Zech et al. (2022b)). In terms of the CAM concept (see Appendix, Section A1), we evaluate the ratio of the number of surface edges of a POM particle next to a pore cell $s_f^p$ and the total number of edges of the POM particle $s^p$. The more a POM particle is occluded, the slower its degradation. A complete occlusion results in the total halt of decomposition, as $s_f^p = 0$. The degradation of necromass is treated likewise. $R_{root}$ describes the input rate of dissolved substrates from root exudation. Finally, a first-order kinetics describes the irreversible loss of dissolved N through leaching $R_{leak}$.

## 2.3 Virtual Soil and Model Parametrization

The parameterization of the model is oriented towards the experimental platforms presented in Vetterlein et al. (2021), where a multitude of studies have been carried out at the same sites or with the same soils to study the spatiotemporal self-organization of the rhizosphere shaped by (maize) plants (Vetterlein et al., 2020). The samples stem from a haplic Phaeozem soil under agricultural use in central Germany. It has been excavated and sieved to provide a consistent, comparable basis for the various experiments. If possible we used parameters from those related studies (Vetterlein et al., 2021; Brax et al., 2020; Niedeggen et al., 2024; Santangeli et al., 2024). We create a virtual soil representing the loam of the soil plot experiment (SPE) which has a bulk density of $1.39 \, \text{g cm}^{-3}$ and a porosity of 0.411 (Vetterlein et al., 2021), which results in a solid density of $2.36 \, \text{g cm}^{-3}$. The particle size distribution of the SPE loam is given in Table 3. Soil particles of realistic shapes are first randomly distributed into the pore space within a domain of size (500 x 500 x 1) $\mu\text{m}^3$ according to the given particle size distribution and porosity. The shapes of the individual particles stem from a library obtained by dynamic image analysis of wet-sieved soil fractions of an arable Cambiosol with a range of clay contents (Schweizer et al. (2019); Figure 4(a)). The amounts of C and N in pools and mass fluxes are often measured relative to the dry soil mass. As the total dry soil mass of our virtual soil can be calculated by the solid density and porosity in the domain, resulting in 0.35 µg soil, experimental data can directly be converted and used for our parametrization. As the SPE loam has been used in studies on microbial growth kinetics for different root-derived substrates (Niedeggen et al., 2024) and the quantification of maize root exudation (Santangeli et al., 2024), we gain further relevant parameters as follows (see Table 4 for all values and references).

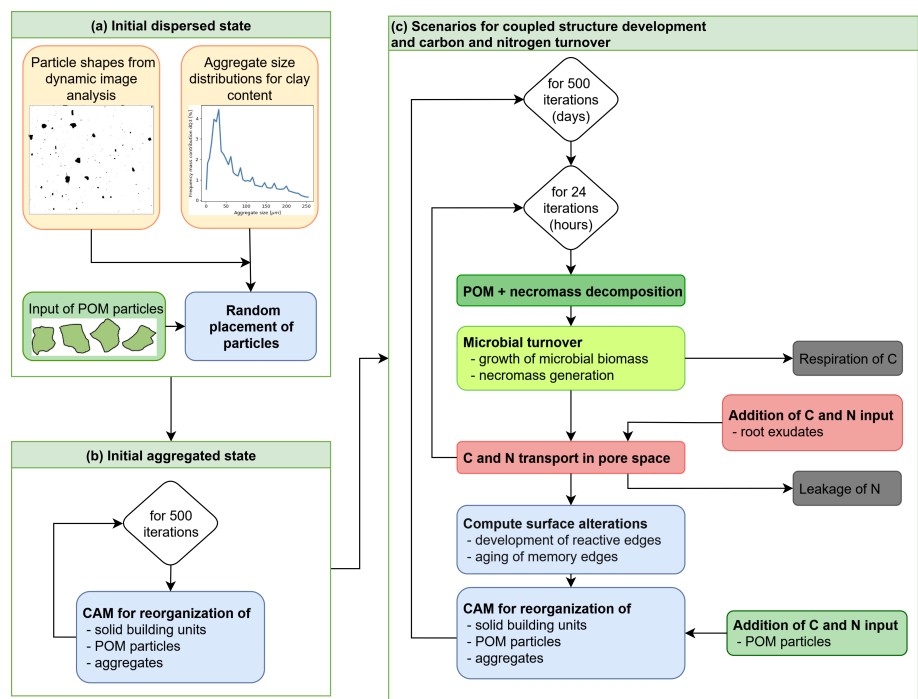

**Figure 4.** Workflow for the simulations: (a) Creation of initial dispersed state, (b) application of CAM to create initial aggregated state, and (c) carbon and nitrogen turnover coupled with structure development in 500-day scenarios.

Throughout the simulations the pore space (with pore sizes between 2 and 150 µm micrometer) is fully saturated with an initial DOC concentration of $C_S^0 = 10^{-5}$ gCcm$^{-3}$ and an initial C/N ratio of 10. On this small scale, C enters the domain either as fine, partially degraded or fragmented POM particles, or as dissolved C from exudates (see below). The C/N ratio of POM was set to 100, which is in the range of wood or root litter (Poeplau et al., 2023). Initially POM particles with a concentration of 0.48 gCcm$^{-3}$, and necromass with a concentration of 0.32 gCcm$^{-3}$ and C/N ratio of 10 amounting to a volume fraction of 5% of the solid area were randomly added to the pore space corresponding to 60 POM and 60 necromass particles each with a size of 6-10 µm in diameter. Given the small scale of the investigated domain, these particle sizes correspond to a fine fraction of POM (Lavallee et al., 2020). This results in an initial amount of 6.24 mgCg$^{-1}$POM-C and 2.61 mgCg$^{-1}$microbial-necromass-C. While up to 70 % of the initial soil C is retained in POM, its contribution to the initial soil N content is 20 %. After this initialization 216 microbial biomass voxels with a density of 63.4 mgCcm$^{-3}$, divided into 72 connected components of 3 cells each are distributed in the pore space. This results in a total amount of microbial biomass of 158 µgCg$^{-1}$, which corresponds to Niedeggen et al. (2024).

In all scenarios we consider a POM input of 0.083 mgCg$^{-1}$ by a random placement of one POM particle with 60 µm$^2$ into the pore space of our computational domain every 10th day (compare low input scenario in Zech et al. (2022b)). The initial amount of total organic carbon (TOC) in the system constitutes 0.89 % of the soil's mass, which is close to 0.85 % as reported in Vetterlein et al. (2021), and in the range of other loamy soils (cf., e.g., 0.74% (Endress et al., 2024)). The

amount of total organic N (TON) is 0.034 % of soils' mass which corresponds to values for crop and grassland soil with low N content; in Vetterlein et al. (2021) 0.083% is reported. We assume that root exudation takes place along a central vertical line of the computational domain (recall the scheme in Fig. 1). For the root C exudation rate we use the values for maize (wild type) in the plant development stages BBCH14 and 19 reported in Santangeli et al. (2024) and for the Monod kinetics of early season exudates we apply the values reported by Niedeggen et al. (2024), see Table 4. Further, Niedeggen et al. (2024) reported an initial microbial biomass $C_B^0$ of 158 µgCg$^{-1}$, a microbial basal activity $R_{\text{basal}}$ of 0.18 µgCO$_2$-Cg$^{-1}$h$^{-1}$ and a peak activity $R_{\text{peak}}$ of 2.54 µgCO$_2$-Cg$^{-1}$h$^{-1}$. Assuming that basal activity is dominated by metabolic activities not related to growth, the maintenance respiration rate can be derived as the fraction between respiration and microbial biomass $k_M = \frac{R_{\text{basal}}}{C_B^0} = 0.3 \cdot 10^{-6}$s$^{-1}$. Similarly assuming that peak respiration is dominated by growth respiration, which depends linearly on the C uptake, the maximal C uptake rate coefficient of a microbial biomass is derived as $\nu_{C_S} = \frac{(R_{\text{peak}} - R_{\text{basal}})/k_G}{C_B^0} = 1.6 \cdot 10^{-5}$ s$^{-1}$. Exudation of a passing root causes a C release with a rate of 3.6 $\frac{\text{µgC}}{\text{h} \cdot \text{cm}^2}$ (Santangeli et al., 2024) in cells aligned to the vertical center line (if the cell is not of type soil particle). One pulse of constant exudation with this rate through the cell surface of 4 µm$^2$ results in a total rise of DOC of 0.01 mgCgsoil$^{-1}$ per day, which is the duration of one pulse in the reference setting.

Based on this model parametrization, the simulations are conducted with the algorithmic workflow illustrated in Fig. 4: Starting from the creation of the initial dispersed state with an initial, random placement of the particles, the subsequent creation of the initial aggregated state by the CAM rules, and then the time loop of the different scenarios including the temporal evolution of the components. While the time step size for the CAM loop is 1 d, for the calculation of reaction and transport terms 1 h is chosen. Details on the CAM and the numerical methods used in the implementation are given in the Appendix, Sections A1 and A3.

## 2.4   Evaluation, Measures

The simulation results are analyzed in terms of the dynamics of the distinct C and N pools (recall Fig. 2). We distinguish the initially present necromass from the one created in the course of the simulation and quantify the relative share of C and N (in %) in the various pools. We also specify the fluxes between them, as defined in Section 2 and depicted in Fig. 2. The Carbon Use Efficiency (CUE) of the microbes is evaluated as ratio between uptake and assimilated carbon(Manzoni et al., 2018):

$$CUE = 1 - \frac{R_O + R_M + R_G}{f_U} . \tag{11}$$

Considering the estimated total dry mass soil in the domain, see Section 2.3, the soil C [mgNg$^{-1}$soil] and N [mgNg$^{-1}$soil] concentrations are calculated for each pool in each timestep and their temporal evolution is depicted. As indicators of potential hotspots, i.e regions of elevated microbial activity or increased concentrations of microbial remnants, driven by processes such as EPS production, root exudation, and exoenzyme-mediated decomposition of POM or necromass, we define the relative availability of dissolved C for the microbes: it is defined by the fraction term $\frac{C_S}{C_S + k_{C_S}} \in [0, 1)$ in the Monod kinetics (Table 2) which indicates portion of the maximum microbial growth can be attained.

**Table 4.** Definition of model parameters and values used in the simulations with references, if available.

| Symbol | Definition | Unit | Value | Reference |
|---|---|---|---|---|
| $\nu_{C_S}$ | Maximum uptake rate | $\mathrm{s^{-1}}$ | $1.6 \cdot 10^{-5}$ | Niedeggen et al. (2024) |
| $kc_S$ | Half saturation constant | $\mathrm{\mu g C g^{-1}}$ | 116 | Niedeggen et al. (2024) |
| $C_B^0$ | Total initial microbial biomass | $\mathrm{\mu g C g^{-1}}$ | 158 | Niedeggen et al. (2024) |
| $C_B^{MIN}$ | Min. microbial biomass C cell conc. | $\mathrm{g C\,cm^{-3}}$ | 0.0016 | Zech et al. (2022a) |
| $C_B^{MAX}$ | Max. microbial biomass C cell conc. | $\mathrm{g C\,cm^{-3}}$ | 0.3168 | Zech et al. (2022a) |
| $N_B^0$ | Initial microbial biomass CAM cells | - | 216 | Zech et al. (2022a) |
| $(C/N)_S^0$ | Initial C/N ratio of the substrates in the soil solution | - | 10 | Langeveld et al. (2020) |
| $C_S^0$ | Initial C concentration in the soil solution | $\mathrm{g C\,cm^{-3}}$ | $10^{-5}$ |  |
| $C^{Ex}$ | Min. C concentration creating a gluing spot | $\mathrm{g C\,cm^{-3}}$ | $10^{-2}$ |  |
| $C_{POM}^0$ | Initial POM cell concentration | $\mathrm{g C\,cm^{-3}}$ | 0.48 | Zech et al. (2022b) |
| $C_{Nec}^0$ | Initial necromass cell concentration | $\mathrm{g C\,cm^{-3}}$ | 0.3168 | Zech et al. (2022b) |
| $k_{POM}$ | Basal POM decay rate constant | $\mathrm{d^{-1}}$ | 0.0096 |  |
| $k_{MN}$ | Basal necromass decay rate constant | $\mathrm{d^{-1}}$ | 0.0096 |  |
| $C_B^k$ | Microbially mediated factor, modifying rates $k_{POM}$ & $k_{MN}$ | $\mathrm{g C\,cm^{-3}}$ | 6.3360 | Kaiser et al. (2014) |
| $D_S$ | Diffusion-dispersion coefficient of dissolved C and N | $\mathrm{cm^2\,s^{-1}}$ | $1.94 \cdot 10^{-10}$ |  |
| $(C/N)_B$ | Constant C/N ratio of the microbial biomass | - | 10 | Miao et al. (2020) |
| $(C/N)_{Nec}$ | Constant C/N ratio of the microbial Necromass | - | 10 | Kaiser et al. (2014) |
| $(C/N)_{POM}$ | Constant C/N ratio of POM | - | 100 | Niedeggen et al. (2024) |
| $(C/N)_{Root}$ | Constant C/N ratio of root exudates | - | 40 | Cochran et al. (1988) |
| $k_G$ | Respiration rate constant for growth and enzyme production |  | 0.26 | Kaiser et al. (2014) |
| $k_M$ | Maintenance respiration rate constant | $\mathrm{s^{-1}}$ | $0.3*10^{-6}$ | Niedeggen et al. (2024) |
| $k_{Nec}$ | Rate constant for microbial biomass turned into necromass | $\mathrm{s^{-1}}$ | $1.6*10^{-6}$ |  |
| $k_{leak}$ | Rate constant for N leakage | $\mathrm{h^{-1}}$ | 0.0001 | Kaiser et al. (2014) |
| $R_{root}$ | Exudation rate of C by root per time and root surface area | $\frac{\mu g C}{h \cdot cm^2}$ | 3.6 | Santangeli et al. (2024) |

## 2.5 Simulation Scenarios

In order to study the C and N dynamics we set up the Scenario DynamicReference (see Section 2.5.1) which is then restricted to Scenario Static by excluding structure dynamics, and extended by varying the quality (Scenario CN) and the amount (Scenario 10Pulses) of root exudates to study the influence of different drivers in the system. The C:N ratio of POM is fixed at 100 for all scenarios. A brief extension analysis varying the POM C:N ratio is provided in the appendix, see Figure A1. All simulations cover 500 days.

### 2.5.1 Scenario 1: C/N dynamics under continuous POM input and a single short-term root exudation

**Scenario DynamicReference: Reference setting**

We consider our complete model outlined in Section 2 and parametrized according to Table 4. Within the total simulation time of 500 days, one pulse of root exudates for one day at a rate of 3.6 $\frac{\mu gC}{h \cdot cm^2}$ and a C/N ratio of 40 is assumed on day 200. This scenario serves as a reference setting.

**Scenario Static: Excluding particle movement**

A key feature of the presented approach is the coupling of structure dynamics to C turnover and vice versa. To investigate the impact of the spatial dynamics in our setting, i.e., the rearrangement of particles, and the coupling of the C and N turnover to the evolution of the pore system, we compare the Scenario DynamicReference - where organic and mineral particles and aggregates relocate over time - to results simulated with a static geometry. For the latter, the initial aggregated state (see Section 2.1) is chosen.

### 2.5.2 Scenario 2: Impact of the quality and amount of exudates

**Scenario CN: Impact of the quality of root exudates**

To study the impact of exudate quality on the C and N cycles, three different C/N ratios are assumed for the single exudation pulse at day 200 which are compared to the Scenario DynamicReference ($C/N = 40$): $C/N = 10$, which is in the range reported for maize mucilage (e.g. Brax et al. (2020)), $C/N = 22$ (Niedeggen et al., 2024) and a C-rich situation is considered by setting $C/N = 100$. The total amount of C exuding is the same in these settings, only the corresponding amount of N varies in the sub-scenarios.

**Scenario 10Pulses: Impact of the amount of root exudates**

In natural soils, multiple fine roots can influence the same soil region within a growth period, or the same root exudes substances in frequent cycles. To mimic such a situation in a simplistic way we establish 10 successive one-day-pulses of root exudation with $C/N = 40$ as described in Scenario DynamicReference. Within a time span of 50 days and at a frequency of 5 days, one day of exudate pulse is followed by 4 days without. The first pulse starts at time $t = 200$ d.

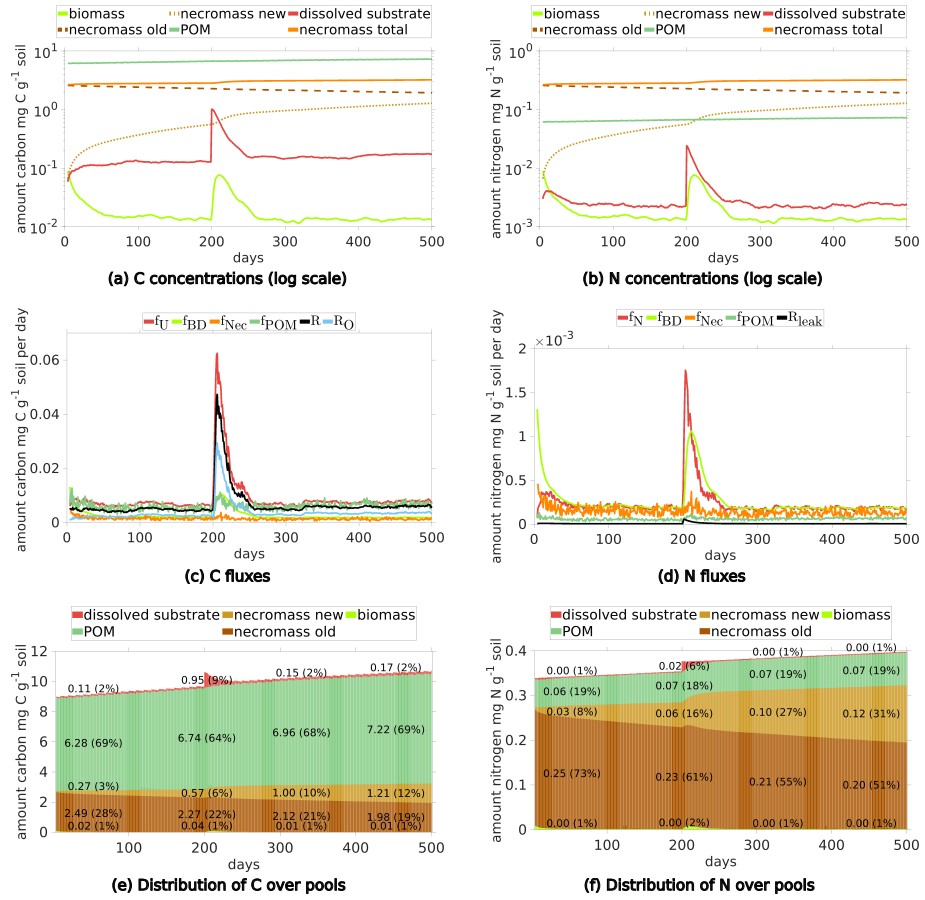

**Figure 5.** Temporal evolution of C and N concentrations ((a) and (b)) and C and N fluxes between different pools (c and d), and the relative distribution of total C and N in different pools ((e) and (f)) for Scenario DynamicReference over 500 days. Continuous input of POM, one exudate pulse (C/N 40) after 200 days. Remember the definition the fluxes and pools, cp. Fig. 2. The concrete values of the distribution in (e) and (f) are depicted at day 50, 200, 325 and 450

## 3 Results

### 3.1 Scenario 1: C and N dynamics under continuous POM input and short-term root exudation

We first depict the results of Scenario DynamicReference (cf. Section 2.5). The temporal development of C and N concentrations, their fluxes and distribution among the different pools (see Fig. 5) reveal three different phases: the initial state up to day 199 (first equilibrium), the time span under the influence of the exudate pulse (days 200 - 250), and the final phase from day 250 up to day 500 (second equilibrium). Additionally, as one of the key insights of our model approach, the localized spatial dynamics of carbon dynamics and hot spots biomass are discussed.

## Initial Phase: Day 1-199

**Dynamics of C and N pools:** The evolution of the C concentrations in the distinct pools are depicted in Fig. 5(a) and the respective shares are illustrated in Fig. 5(e). We observe the increase of POM due to its continuous addition (dark green curve), the slow degradation of initially present necromass (dark brown) and the generation of new necromass over time (light brown) due to the death of living biomass. Departing from the initial state the C concentrations in the soil solution (red curve), the total necromass (orange curve) and – after approximately 75 days – also the microbial biomass (light green curve) tend to dynamic equilibria. The N concentrations show the corresponding behavior (Fig. 5(b) and Fig. 5(f)).

Under the impact of continuous POM input and turnover of OM, the size of the microbial biomass reaches its dynamic equilibrium of 0.015 mgCg$^{-1}$soil after around 75 days. As such the biomass makes up less then 1 % of the total SOC (see Fig. 5(e)), which is common for agricultural soils (Wei et al., 2022). Likewise, necromass decomposition and its simultaneous formation through the decay of microbial biomass result in a dynamic equilibrium of approximately 2.8 mgCg$^{-1}$soil, which is within the range of the initial amount of necromass. As illustrated in Fig. 5(e), this sums up to a fraction of around 30 % of SOC, but around 80 % of total N (Fig. 5(f)) (e.g. Zhang et al. (2022) report 40% of SOC). After a short initial increase of the dissolved C and N concentrations (cf. the red curves in Fig. 5(a) and 5(b)), they stabilize at approximately 0.12 mgCg$^{-1}$soil and 0.0025 mgNg$^{-1}$soil after 25 days, which is in the range of lab experiments, see, e.g. (Cookson et al., 2007). So, the soil solution contains a C/N ratio between 40 and 55.

**Fluxes:** The dynamic equilibria of the state variables are also evident from the fluxes as depicted in Figures 5(c) and 5(d). Up to the simulation time of 200 days the fluxes are varying around their individual equilibria, where the decomposition rates of POM $f_{POM}$ and necromass $f_{Nec}$ exhibit the strongest fluctuations. The similar magnitude of the influx i.e. the turnover of biomass into necromass $f_{BD}$ and the outflux, i.e. the necromass decay $f_{Nec}$, results in a dynamic equilibrium of the total necromass. Likewise, the uptake of dissolved C is equal to the sum of respiration and microbial decay. In our simulation, the mean of the total respiration between day 75 and 199 is around 4.8 µgCO$_2$-Cg$^{-1}$day$^{-1}$, which aligns with Niedeggen et al. (2024), where a basal microbial activity of 0.18 µgCO$_2$-Cg$^{-1}$h$^{-1}$ = 4.32 µgCO$_2$-Cg$^{-1}$day$^{-1}$ was measured. The respiration overflow (due to the imbalance of C/N for biomass) accounts for about 55 % of the total respiration. Concerning N, the equilibrium in the system is reflected by the balance between assimilation $f_N$ and loss through microbial decay $f_{BD}$. The high N flux of $f_{BD}$ during the first 20 days (light green, Fig. 5(d)) corresponds to the high initial degradation of biomass.

## Influence of exudate pulse: days 200 - 250

**Pools:** After 200 days, a flush of root exudates with C/N 40 enters the system via the soil solution along a central line through the domain (see the description of Scenario DynamicReference in Sec. 2.5). As a consequence, a rapid increase of dissolved C (red peak in Fig. 5(a)) and of dissolved N (red peak in Fig. 5(b)) is observed. The nutrient availability results in a subsequent increase of the microbial biomass (light green in Figs. 5(a) and 5(b)), followed by a subsequent, modest increase of new microbial necromass (light brown). The peak of the microbial biomass is reached 11 days after the pulse. The N concentrations (see Fig. 5(b)) exhibit the same behavior as the C concentrations, but at different absolute levels.

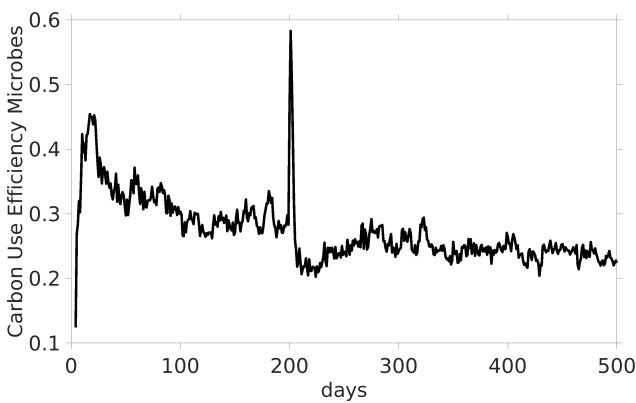

**Figure 6.** CUE of the ~~reference~~ Scenario DynamicReference over time.

**Fluxes:** Most evident are the high uptake fluxes of DOC and dissolved N (red curves $f_U$ and $f_N$ in Figs. 5(c) and 5(d), respectively) which enable the subsequent microbial biomass growth. In the sequel the decomposition of microbial necromass and into dissolved carbon have doubled, while the microbial biomass decay flux is even 5.6 times higher than before the exudation pulse. The peak activity of the microbial biomass leads to a total respiration of 47 μgCO$_2$-Cg$^{-1}$day$^{-1}$ cf. black curve in Fig. 5(c), compared to Niedeggen et al. (2024), who reported a peak of 2.54 μgCO$_2$-Cg$^{-1}$h$^{-1}$ = 60.96 μgCO$_2$-Cg$^{-1}$day$^{-1}$. The increase of dissolved nitrogen entails a tenfold increase in N leakage. Further studying the N fluxes (namely the assimilation/mineralization $f_N$ and the biomass degradation $f_{BD}$) the biomass dynamics can be reconstructed: if $f_N \approx f_{BD}$ the microbial biomass stays in equilibrium (before the pulse). As the pulse enters the system $f_N >> f_{BD}$ and the microbial biomass grows for 8 days. Then $f_N < f_{BD}$, and the microbial biomass degrades again.

**Second equilibrium: days 250 - 500**

The same equilibria for DOC and dissolved N as before the exudate pulse are reached within 50 days, and with a delay of a few days also for the microbial biomass (see Figs. 5(a) and 5(b)). The main difference between pre- and post-pulse situation is the increase in microbial necromass (orange curves) while POM rises slightly. In total, this corresponds to a increase in TOC to 1.072 % of the soil's mass. Also, all fluxes have returned to their equilibrium values (compare Fig. 5 before day 200 and after 260 days). Up to fluctuations, the equilibrium values remain constant until the end of the simulation time.

**CUE and localized spatial dynamics**

Finally, we present the temporal dynamics of the CUE of microbes. For the dynamic equilibrium between day 100 and 200 there is a CUE with a mean of 0.289. The CUE immediately reacts on the exudate pulse on day 200 and its peak is obtained on day 201 with a value of 0.583, see Fig. 6. As the main feature of our model is its capability to resolve localized processes, we illustrate the spatially explicit heterogeneous distribution of biomass and necromass in the domain along with the C availability exemplarily at three time steps in Fig. 7. As the connectivity in the pore space essentially determines the local availability of

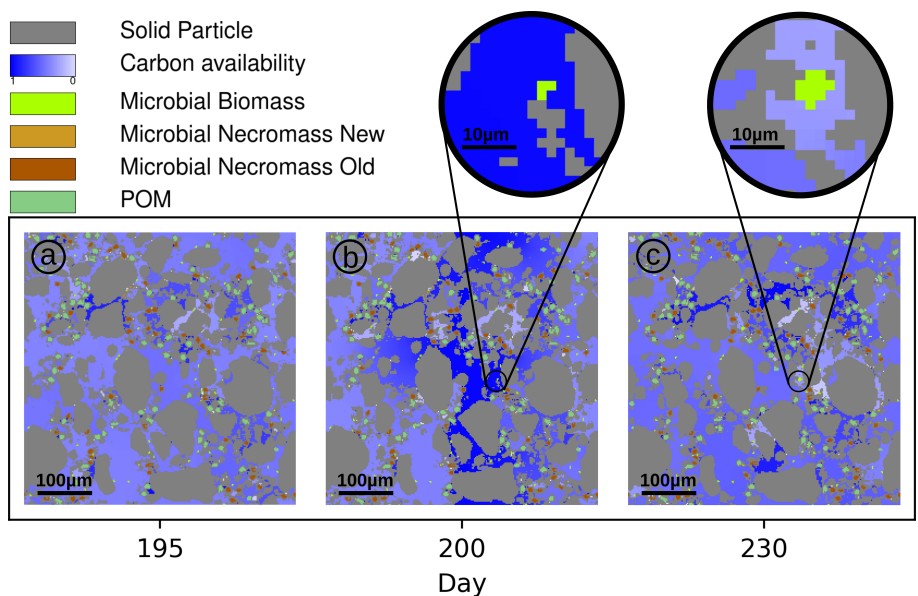

**Figure 7.** C availability (normalized; dark blue = high, light blue = low) in pore space in Scen. DynamicReference at days 195 (a), 200 (exudate pulse, (b)) and 230 (c). The lenses highlight a hot spot of biomass growth after the pulse.

dissolved C and N in the domain (Zech et al., 2022a), this results in locally variable microbial growth. If both substrate and
consumer (microbes) are connected in small but isolated pores we observe that C is depleted, i.e. the availability (see definition
above) is less than 0.4 (light blue regions in Fig. 7). In large, connected pore spaces, the amount of dissolved C available to
the biomass is up to 0.7. Note that degradation of POM and microbial necromass also generates new DOC. On the other hand,
in pores that are disconnected from living biomass, the C availability remains high, with values up to 1.0 (dark blue zones in
Fig. 7). As we explicitly take into account the restructuring of the pore space, the connectivity may dynamically change. At day
200 C-rich exudates enter the center of the domain (dark blue regions of Fig. 7(b)). Connected regions near the source of root
exudation facilitate high C uptake rates by microbial biomass, leading to an increase in microbial growth (example: Figure 7,
black circles). The microbial community grows in that hot spot under favorable conditions between day 200 (Fig. 7(b)) and 230
(Fig. 7(c)). In contrast, disconnectedness, i.e. physical barriers may prevent substrate exchange and so, new C-rich pores may
persist after the exudate pulse because no consumers have access to them. As such the microbial dynamics are significantly
impacted by structural changes and locally highly diverse.

## 3.2 Scenario Static: Impact of structure dynamics

We compare the Scenario DynamicReference to Scenario Static, where the structural rearrangement is disregarded to illustrate
the impact of altering soil structures on the carbon and nitrogen pools and fluxes as well as spatially localized. The evolutions
of the C and N concentrations are depicted in Fig. 8(a) and (b), where we focus on the most dynamic phase from day 180 to
350. We observe two notable differences in the Scenario Static: The increased concentrations of dissolved C and N (broken red

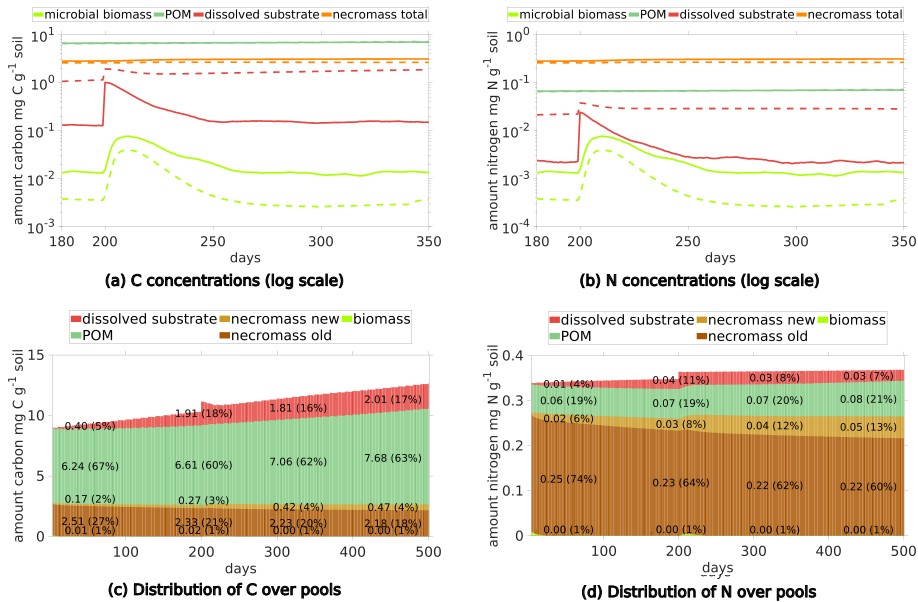

**Figure 8.** Temporal evolution of C and N concentrations (a) and (b) in Scenario DynamicReference (solid lines) and Scenario Static without particle movement (dashed lines). Relative distribution of total C (c) and N (d) in different pools in Scenario Static without particle movement

.

lines in Fig. 8(a) and 8(b)), and at the same time the biomass concentrations exhibit a lower level (broken light green lines), e.g. C in microbial biomass is only 30 % of the amount of C in biomass in Scenario DynamicReference. Ten days after the exudate pulse the microbial biomass peaks at 0.039 mgCg$^{-1}$soil, which is 49 % less compared to the scenario with particle movement. After 250 days, the effect of the pulse has vanished and dissolved N decreases slightly, while dissolved C continues to increase, but with less incline than before the pulse. In general the amount of dissolved carbon in Scenario Static is up to 8 times higher compared to Scenario DynamicReference, dissolved nitrogen even up to 10 times higher. The distributions of C and N in the pools in Fig. 8(c) and 8(d) underline the effect of structure dynamics if compared to Figs. 5(e) and 5(f)). More DOC and dissolved N remains in solution in the static scenario, less active biomass and less necromass is generated due to its limited access to the available nutrients. Without movement of the particles TOC increases to 1.26 % of the soil's mass at the end of simulation. The difference of 1.96 mgCg$^{-1}$soil between both Scenarios, DynamicReference and Static, is a consequence of the lower $CO_2$ respiration in Scenario Static. Furthermore it can be noted that the dissolved substrates and microbial biomass are fluctuating more in Scenario DynamicReference, which is a direct consequence of the break-up of structures / bindings among the particles and thus changing connectivity between POM, necromass and biomass. For illustrating these mechanisms, Figure 9 shows two snapshots at day 130 and 135 visualizing the spatial distribution of the C/N ratio in the soil solution. While the C/N distributions of the static geometries are almost identical ((c) and (d), Scen. Static), the C/N distribution changes in the Scenario DynamicReference ((a) and (b)). The circled area points to the break-up of a structure that isolates two regions with high and low C/N at day 130. At day 135 the rearrangement of particles connected the two pore regions and the C and N

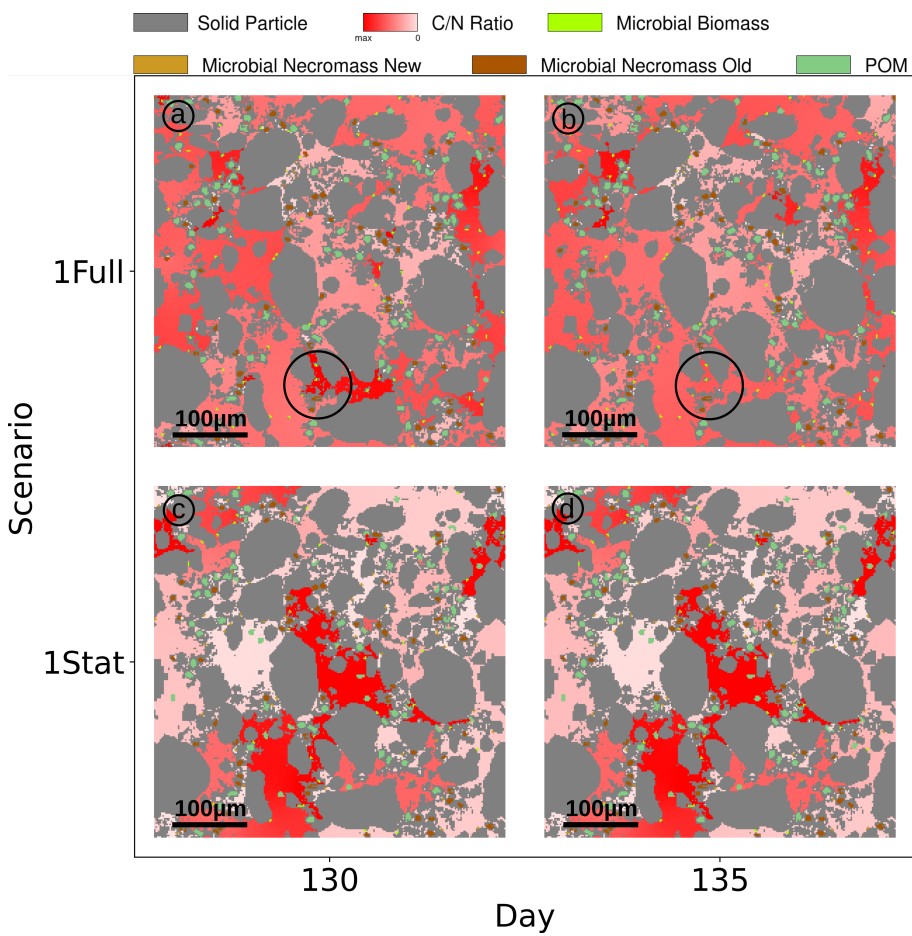

**Figure 9.** C/N ratio of soil solution at days 130 and 135 for the full Scenario DynamicReference ((a) and (b)) and without particle dynamics Static ((c) and (d)). Circle indicates exemplarily where pores get connected by particle movement.

concentrations equilibrated. In general, the difference of the C/N ratios between unconnected pores is higher in Scenario Static, leading to a pronounced patchiness. The amount of dissolved C and N in the pores is depending on the connectivity: In pores that are separated from spots with microbial biomass by physical barriers the dissolved C accumulates and exhibits a high potential carbon availability and C/N ratio. In contrast, in isolated pores containing microbial biomass, C and N are consumed, but without fresh C and N sources the C/N ratio in solution can increase while the uptake decreases until the biomass is not able to fulfill its basal activity and degrades. As such the impact of soil restructuring on carbon and nitrogen dynamics is evident, compare also the summarized final states in Figure 11.

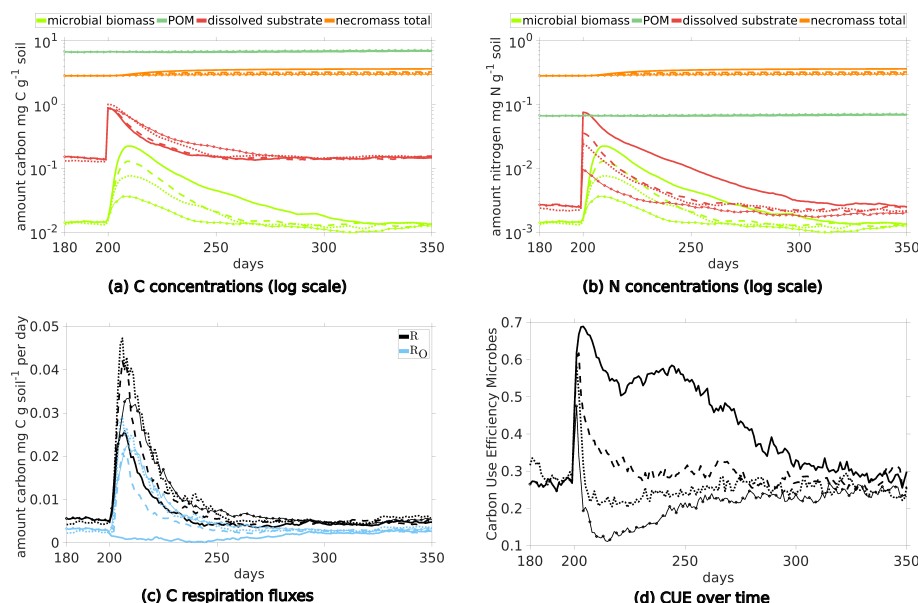

**Figure 10.** Temporal Evolution of C and N concentrations ((a) and (b)) and C respiration fluxes (c), and carbon use efficiency (d) for sub-scenarios comparing the impact of the quality of exudates: One pulse at day 200 with C/N ratios 10 (straight lines), 22 (dashed), 40 (dotted; Scen. DynamicReference) and 100 (straight lines with circles).

## 3.3 Scenario CN: Impact of the quality of root exudates

The alteration of the C/N ratio (10/22/40/100) of the root exudates does not influence the evolution of the concentrations until the exudation pulse, so the same equilibria as in Scenario DynamicReference are established for all pools (see Fig. 10(a) and 10(b)). Small differences in the concentrations are due to the random break-up and placement of POM which may differ. Investigating now the impact of the exudation pulse, altering the C/N ratio of root exudates does not affect their carbon mass, resulting in similar dissolved C peaks (red lines in Fig. 10(a)). Dissolved N decreases according to the given C/N ratio 10/22/40/100 in the sub-scenario (Fig. 10(b)). Higher amounts of dissolved N foster microbial growth, and accordingly C in biomass peaks are higher (light green curves in Fig. 10(a)) and the elevated level persists longer until the previous equilibrium state is reached after the pulse. For a pulse with C/N ratio of 10 it takes 125 days, while for a C/N ratio of 100, less biomass is produced and the pre-pulse level is attained already after 50 days.

Studying the CUE (see Fig. 10(d)), it is evident that there is a different response for the exudates with C/N ratio of 10 and 22 compared to 40 and 100: All sub-scenarios exhibit a rapid increase in the CUE with peaks of 0.68 for C/N 10, 0.61 for C/N 22, 0.58 for C/N 40, and 0.47 for C/N 100, but their evolution differs subsequently: After the peak, CUE for C/N 40 and 100 briefly drops below their pre-pulse level of 0.3, recovering 50 and 100 days after the pulse, respectively. For C/N 22, the CUE drops to the pre-pulse state 30 days later, while for C/N 10 the microbial biomass maintains a CUE around 0.5 for a time span of almost 100 days. Remark that a higher C/N ratio in root exudates corresponds to a prolonged elevated respiration (Fig. 10(c)

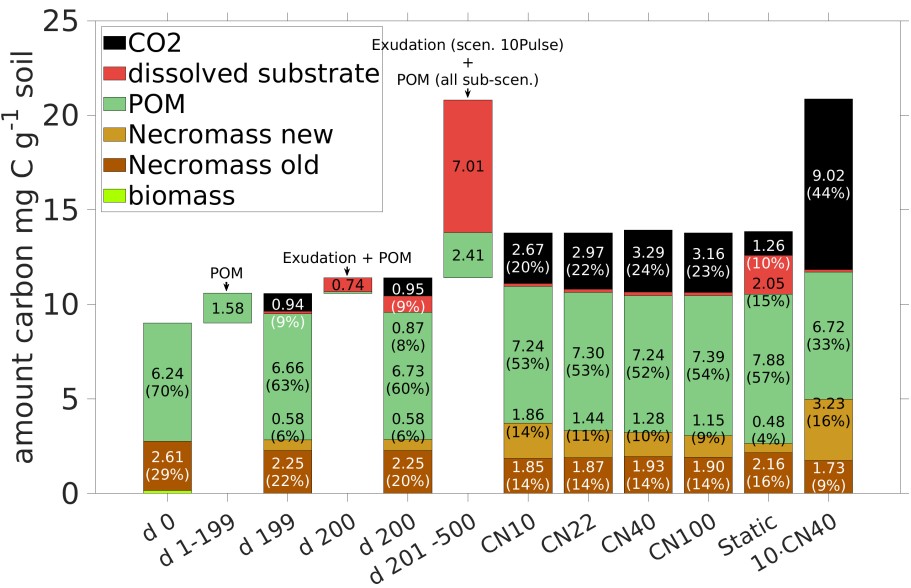

**Figure 11.** Total amount of C and distribution among the pools at time steps 0, 199, 200 (exudate pulse) and 500 is compared for different exudate qualities (C/N 10/22/40/100) in Scenario CN and for the 10pulse-Scenario 10Pulses (C/N 40) (last bar). Additionally the amounts of C entering the system are displayed (POM and exudates) for the period 1-199 (POM, all scenarios), day 200 (exudates and POM, all scenarios), and days 201-500 (exudation in secenario 10Pulses, POM for all scenarios). For clarity, the values of pools contributing less than 2 % (biomass and dissolved substrate except in Scenario Static) are not shown.

straight lines with circles and dotted lines), leading to a larger total loss in $CO_2$. The total respiration peak $R$ caused by a root exudation pulse with C/N ratio 10 (solid lines) exhibits the lowest among the different sub-scenarios. Strong differences are apparent comparing the respiration overflow of C due to the imbalance in C/N ratio in the biomass (light blue lines in Fig. 10(c)): While there is almost no C overflow when exudates had C/N ratio 10, there is C overflow in all other sub-scenarios.

As such a change in the root exudates' C/N ratio significantly impacts CUE, necromass formation, and C respiration.

The total amounts of added and lost C, and its distribution in the system at specific time points are summarized in Fig. 11. The microbial response to the different exudate qualities is indicated by the creation of new necromass during the simulations, as it results from the total amount of biomass ever existed. Comparison of day 200 with day 500 in Fig. 11 demonstrates and quantifies that higher C/N ratios in exudates lead to lower necromass accumulation but greater $CO_2$ release. From day 200 until

390 day 500 in total 1.28 mgCg$^{-1}$soil necromass is generated from biomass if the pulse had a C/N ratio of 10, and simultaneously 1.72 mgC-$CO_2$ gsoil$^{-1}$ is released. On the other end, for a pulse with C/N 100 only 0.57 mgCg$^{-1}$soil necromass is generated, while 2.31 mgCg$^{-1}$soil is respired. Recall also the Scenario Static which is included in Fig. 11 for comparison, showing significantly less $CO_2$ and necromass production, along with higher DOC concentrations.

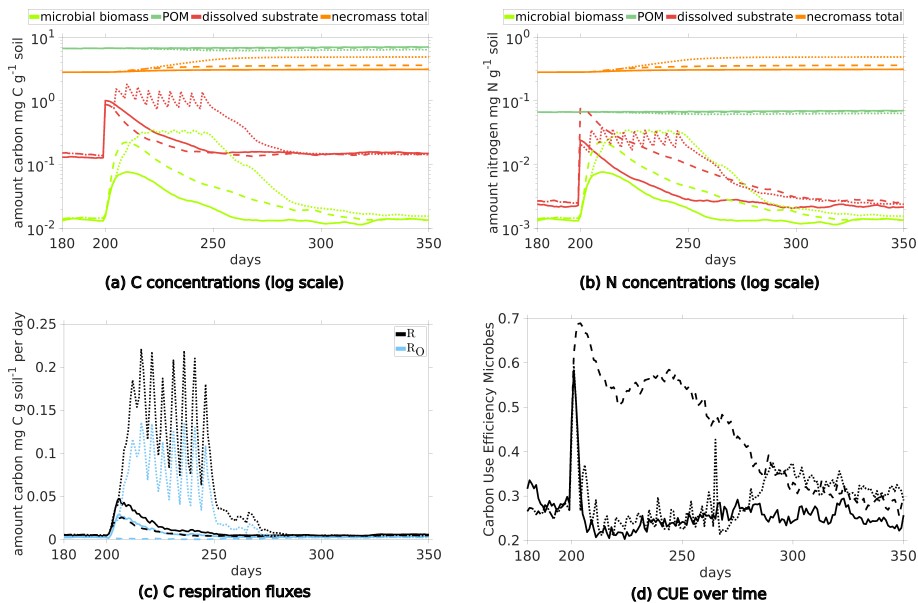

**Figure 12.** Temporal Evolution of C and N concentrations ((a) and (b)) and C respiration fluxes (c), and carbon use efficiency (d) for scenarios comparing the impact of the amount of exudates: Comparison of one pulse (at day 200) scenario with CN40 (straight lines, Scenario DynamicReference) and CN10 (dashed, sub-scenario of CN) to 10 pulses with C/N 40 from day 200 to 250 (dotted).

### 3.4 Scenario 10Pulses: Impact of the amount of root exudates

The ten pulses (with C/N 40) over 50 days increase dissolved C and N levels (red dotted curves in Figs. 12(a) and 12(b)). In response, the biomass also grows longer (dotted light green curve in Figs. 12(a)) compared to any scenario with one pulse (DynamicReference or sub-scenario of CN with ratio C/N 10). The maximum biomass concentration following a single pulse with C/N 10 is of similar height, while the biomass peak induced by a single pulse with C/N 40 is at most $\frac{1}{3}$ of the other maxima. Consequently, more necromass accumulates, resulting in a higher TOC of 1.18 % by the end of the simulation. Notably, ten

pulses result in a necromass of 3.23 mgCg$^{-1}$soil and a $CO_2$ production of 9.02 mgCO$_2$-Cg$^{-1}$soil in total, compared with one pulse with high quality with a necromass increase of 73 % to the cost of 238 % more respiration. This is illustrated by the comparison of the bar plots of the final states in Fig. 11 and also expressed by a low CUE (Fig. 12(d)). Although the amount of C and N from ten exudate pulses now is ten times higher than in the Scenario DynamicReference, a high share is respired. Comparing the distribution at day 500, we observe that the amount of C in the pools sums up to 56 %, while 44 % has been

respired. For the single pulse with C/N 40, only 24 % of the C have been transformed to $CO_2$ while 76 % are still stored in the domain (Fig. 11). We note that among all our presented scenarios with structure dynamics, one pulse with C/N 10 produces least $CO_2$ (in total 2.67 mgCO$_2$-Cg$^{-1}$soil, 20 %) and stores second most C (summing up to 80 % and 11.1 mgCg$^{-1}$soil) . Note also the differences in the POM contents at day 500 in Fig. 11: Although the same amount of POM enters the system in all simulations, the least amount is present finally in the Scenario 10Pulses (6.72 mgCg$^{-1}$soil). This is a direct consequence of the

higher turnover of the living biomass, initiated by the high amount of exudates and resulting in the highest amount of newly generated necromass. Over all simulations, Scenario Static shows clearly highest DOC (15%) and least new necromass (4 %) along with lowest $CO_2$ ouput (1.26 $mgCO_2\text{-}Cg^{-1}$soil).

## 4 Discussion

Discussing the results presented in Section 3 we highlight three key insights: the role of soil structure and local connectivity,
nitrogen limitation, and on the spatial context of accumulation of microbial necromass.

### 4.1 The importance of heterogeneity and connectivity

Under the given model assumptions our study underlines that an evolving physical structure plays a critical role for C turnover (Lucas et al., 2021). In particular, the fully dynamic forward simulations (DynamicReference) including a mechanistic view on OM turnover without inverse parameter fitting reveal the emergence of equilibria in the different C and N pools, as ob-
420 served in natural soils (Lehmann and Kleber, 2015). The connectivity of nutrient sources and consumers, provided by the soil solution, alters due to the dynamic rearrangement of pores and solid matter, defining microenvironments with distinct locally varying concentrations of dissolved organic matter and biomass (see Fig. 7 and 9). Microbial biomass patches residing in pores directly connected to sources of root exudates or decaying organic material benefit from favorable conditions promoting localized growth. In contrast, microbial biomass in isolated regions without direct access to substrate—due to limited
pore connectivity—experience resource limitation and consequently decay. Thus, local (dis-)connectivity can inhibit access and consumption, resulting in the observed increase by 5 to 10 times of dissolved C and N, and simultaneously 30 % lower biomass concentrations in the static Scenario Static (Fig. 8). This effect is also evident for root exudates as their deposition is restricted by soil structure and thus accumulates near the root within an average distance of about 100 μm, see Fig. 7 day 200. This localized enrichment of C and N creates hotspots for biomass growth, highlighting how physical barriers shape nutrient
distribution and microbial growth in the rhizosphere. Furthermore, the addition of labile C and growth of biomass initiates gluing spot evolution and thus has an effect on structure dynamics, resulting in a feedback loop between microbial biomass and soil properties  (Philippot et al., 2024). Already in Zech et al. (2022b) it was pointed out that higher organic matter input (there in terms of POM) led to larger aggregates which was mainly due to increased amount of organo-mineral associations. These local concentration differences are highly sensitive to structural disruptions. Continuous (dis-)aggregation processes
trigger the (de-)occlusion of POM and necromass, leading to constantly changing localized availability of C and N due to their accessibility-dependent decomposition (see in Fig. 7 and 9). Simultaneously, the restructuring of the soil matrix causes continuous changes in pore connectivity. Together, these two local factors - substrate accessibility and connectivity - govern the spatial distribution and availability of substrates, resulting in a dynamic rather than static equilibrium of microbial biomass (Fig. 8). Static geometries lack an important aspect mimicking the dynamic equilibria encountered in natural soils (Lehmann
and Kleber, 2015).

## 4.2 Nitrogen limitation outweighs carbon surplus

By extending the C turnover model respecting the N requirements of the microbes we were able to trace and quantify the interplay between the C and N concentrations and biomass development in the context of a loamy, agricultural soil (Vetterlein et al., 2021). The turnover of living microbial biomass into necromass was also included and provides an important pool of substrates (Liang et al., 2019), along with POM. The system behavior during root exudation of maize plants was studied. All simulation scenarios confirm that microbial biomass dynamically adapts its size to the locally available nutrients (dissolved C and N). The C uptake rate is primarily driven by the amount of DOC while the maximum possible growth rate is constrained by the amount of the available N (Schimel and Weintraub, 2003). We observed that, when a continuous supply of dissolved substrate is maintained —via successive degradation of necromass and POM, along with the regular addition of fresh POM particles — consequently the microbial biomass stabilizes at a certain level. The stabilization occurs independently of the current biomass' size as e.g. defined by the initial conditions or short-term fluctuations due to root exudation. This underlines that microbial biomass dynamics are primarily driven by the current availability of C and N (Kaiser et al., 2014). The persistent overflow respiration of microbial biomass observed across all scenarios is a model-based response to address the imbalance for the microbial metabolism due to a lack of N. N losses from biomass through its transformation into necromass can only be compensated by N uptake to a specific extent, underlined by the dynamic equilibria of fluxes. In contrast, C losses due to maintenance respiration and necromass formation can always be rebalanced, the microbial biomass responds with elevated respiration rates to the surplus of C, and to counterbalance the nitrogen limitation hampering growth. As a substantial portion of the organic matter present in the system exists as POM, with a C/N ratio ten times higher than that required by microbial biomass, this results in the surplus of C. In scenarios, where the C/N ratio of the exudates is closer to 10, biomass growth was stimulated. Under such conditions, overflow respiration was markedly reduced, resulting in an considerably increased CUE, see also Manzoni (2017). As such, in the scenario with the lowest C/N ratio of 10 ( Scenario CN), a temporary nitrogen saturation suggests that microbial biomass can adapt to root exudation dynamics, achieving the highest CUE.

A comparison of peak respiration following root exudate input under different C/N ratios suggests that no direct conclusions can be drawn regarding C assimilation or CUE based solely on these peak values. A peak respiration of $28\pm2.5$ µgCO$_2$-Cg$^{-1}$day$^{-1}$ could be observed in two distinct simulation scenarios: CN sub-scenario with C/N ratio 100 involves a smaller microbial biomass exhibiting a high overflow respiration, while the sub-scenario with C/N ratio 10 exhibits a larger biomass and maintaining a low overflow respiration. Efficient C utilization becomes apparent only over longer time frames, during which more necromass accumulates. This finding suggests that comparison between the initial and final system states provides more meaningful insights into microbial CUE and nitrogen limitation than a single-point respiration peak measurement. A single one-day exudation pulse with a favorable C/N of 10 elevated CUE more than 100 days, while 10 peaks with C/N 40 (10Pulses) could not increase CUE (Fig. 12). On the other hand, the ten exudate pulses triggered the growth of living biomass, and subsequently led to the formation of most necromass, but also to the highest degradation of POM among all scenarios. These results emphasize the importance of the mechanistic model approach to explore the dynamics behind CUE, beyond fixed stoichiometric relations among pools of carbon.

## 4.3 Long-term C and N storage by means of microbial necromass

Due to the low N content of POM, the N required for biomass growth is primarily recycled from necromass. As a result, the overall accumulation of necromass stagnates or progresses only slowly. However, surplus C from POM efficiently supports microbial respiration, allowing the microbial community to maintain activity even under N-limited conditions. During short term root exudation pulses, additional N becomes available and is rapidly taken up by the microbial biomass. A portion of this biomass subsequently decays into necromass, effectively transforming readily available organic material into a more stable, slowly degradable form. Consequently, necromass accumulates and is incorporated into the soil matrix, contributing to long-term C and N storage (Buckeridge et al., 2022). This dynamics results in a net increase in necromass following the exudation event, with highest amounts of resulting necromass for lowest C/N ratios of the exudates (14 % of TOC for C/N ratio 10). In our virtual loamy soils, POM and necromass can become physically occluded within fine soil particles. Without structural disruption, much of this organic matter remains protected from microbial decomposition, extending its residence time in the soil (cf. Zech et al. (2022b)). This is supported by the results of Scenario Static (without structure dynamics), where both microbial biomass and respiration are significantly reduced compared to the Scenario DynamicReference.

## 4.4 Model extensions

Future work could incorporate the dynamics of phosphorous compounds in a similar manner and thus study their importance for microbial growth in particular for agricultural soils (Kwabiah et al., 2003). Fungal bio/necromass constitutes another important C and N pool in soils (Liang et al., 2019) which can be treated in an analogous way. The effects of environmental factors as temperature or moisture variations are not investigated in this study, but could give rise to further extensions.

## 5 Conclusions

The presented approach provides novel insights into the microbially mediated processes that govern the spatial and temporal dynamics of carbon and nitrogen in soils at the micrometer scale. Rather than relying on models including fitted parameters, we employed forward simulations based on assumptions and parameter values directly derived from field and laboratory studies in a comparable context, allowing us to investigate the underlying drivers of observed phenomena in a mechanistic and transferable way. The results are coherent with several experimental findings and support that the model incorporates the relevant mechanisms appropriately. Thus simulations allow us to analyze the distinct responses of the soil-microbe-carbon system to the input of slow-cycling or fast-turnover carbon. As emphasized by Liang et al. (2019), next-generation field management needs to stimulate microbial biomass formation and necromass preservation to maintain healthy soils. The presented simulations support that their respective growth depends predominantly on the favorable C/N ratio of exudates rather than on a large amount, i.e. exudate composition has been shown to have an important impact on C stability. Conversely, in soils that are subject to frequent structural changes microbial activity is notably higher due to the regular exposure and decomposition of

505 previously inaccessible organic matter. This increased accessibility enhances microbial biomass formation, leading to higher elevated $CO_2$ emissions.

*Code and data availability.*  The data and code supporting the findings of this study are not publicly available, but are available from the corresponding author upon reasonable request.

## Appendix A: Supplementary Information

### A1  Cellular Automaton Method

For the Cellular Automaton Method (CAM), a computational domain (here: $(500 \times 500 \times 1)\mu m^3$) is discretized into cells (here: voxels of size $2\,\mu m \times 2\,\mu m \times 1\mu m$), which are the smallest entities of the model. A specific type or state is assigned to each of these cells: solid, pore, POM, biomass or necromass. Several cells may be combined to larger entities to mimic a primary mineral soil, POM or necromass particle or micro-aggregate. While soil, POM, necromass, and pore cells have exclusive states, 515 biomass cells may include both living bacteria and necromass due to the continuous turnover of living bacterial biomass into necromass. Pores as well as biomass cells may additionally contain dissolved substrates such as DOC and dissolved N.

The basic principle of a CAM is the application of rules for the potential relocation or transformation of cells in every time step. In our model, this includes the movement of solid particles, the turnover of POM, the evolution of biomass and necromass, the exudation and distribution of dissolved C and N within certain CAM cells, and changing surface properties of the edges of 520 the solid cells. In every time step, primary mineral soil, POM and necromass particles as well as aggregates thereof, relocate to achieve the most stable configuration, for details see Ray et al. (2017); Rötzer et al. (2023); Rupp et al. (2018b); Zech et al. (2022b). The range of potential movement per timestep (day) increases with decreasing size as small particles are more mobile than large particles/aggregates, see Zech et al. (2022b) for details. It varies from 10 µm for units smaller than 100 $\mu m^2$ over 2 µm for units between 1600 and 80,000 $\mu m^2$, to larger aggregates, which remain immobile. The attractiveness of spots within 525 the potential range of movement and thus the stability of bindings is ranked by the contact types: The connection between permanent reactive surfaces of primary soil particles or aggregates is considered the least attractive, while organo-mineral bindings between POM or necromass particles and reactive surfaces of soil particles are more favorable, with smaller soil particles having a higher proportion of such reactive surfaces. The most attractive configuration is between biomass, POM or necromass particles and primary soil particles or aggregates via so-called gluing spots. The latter evolves at the edge between 530 a solid cell and pore space cell, where the combined amount of C in the DOC and microbial biomass exceeds a threshold $C^{EX}$. As outlined in Zech et al. (2022b) gluing spots undergo an aging process over time in the absence OM and turn into so called memory edges. Ultimately, this aging results in the loss of their surface-binding capacity and therefore stability of the respective binding. Finally, a random break-up of bindings is included to implicitly represent mechanical stresses due to, e.g., movement root growth or earthworms. The more stable the binding, the less probable the break-up. Besides these relocation 535 mechanisms, we particularly include the transformation of biomass to our CAM. Although we assume that the living microbial

biomass is immobile, they can spread by growth to neighboring pore or biomass cells once the carrying capacity of a biomass cell, i.e., the maximum mass of living or dead bacterial cells it can contain, is reached. This concept is adapted from Portell et al. (2018); Zech et al. (2020) and scaled accordingly to our situation. The corresponding maximal C concentration $C_B^{MAX}$ of the microbial biomass is reduced by half of the necromass concentration $C_{Nec}$ in each CAM cell, assuming that the fragments of two dead bacterial cells, which no longer store water, occupy the space of one living cell. The corresponding amount of N in the microbial biomass cells is redistributed likewise to maintain the C/N ratio. On the other hand, if the microbial biomass concentration in a cell $C_B$ falls below a certain threshold $C_B^{MIN}$, equivalent to the minimum C concentration required for a single bacterial cell according to Portell et al. (2018), the cell state transitions from biomass to pore and the remaining C and N are released as DOC and dissolved N the soil solution. If all microbial biomass is fully converted into necromass, ~~the state is exclusive and~~ the same rules and principles apply for each necromass cell as for POM, i.e. it can relocate and decompose (details in section 2.2 in main document).

## A2  Initial aggregated state

To create an aggregated initial state after the initial random placement of particles, the solid building units, POM and necromass particles are relocated within our cellular automaton framework due to their mutual attraction over 500 steps (without reactions taking place, see also the work flow in Fig. 4).

## A3  Implementation

The implementation and simulations of the combined CAM and differential equations model as described in Section A1 build upon Rupp et al. (2018b) and Zech et al. (2022b) and were realized in Matlab (Inc., 2024). The relocation of soil, POM, and necromass particles and aggregates are performed in the CAM framework within a time step of one day. For each such timestep and for each pore, biomass, POM and necromass CAM cell, all biochemical transformations and stoichiometry are calculated, followed by the redistribution of the microbial biomass. These processes are resolved in time with a timestep of one hour to account for the rapid reaction rates. To account for the diffusive as well as reactive components of our model, a non-iterative splitting is employed. More precisely, the solution of the diffusion equation is not iteratively feedbacked into the mass balance equation. Disregarding the diffusive parts, the corresponding system of ordinary differential equations (1), (2), (5)–(9) is solved for each voxel with explicit Runge-Kutta scheme (ode45 in Matlab) for all state variables (compare Table 1 in main document). Thereafter, these solutions are employed as reactive term and the remaining diffusive problem, which is discretized by a Local Discontinuous Galerkin method and implicit Euler, is solved (Rupp et al., 2018a). The procedure is repeated 24 times until the larger time step of one day is reached.

## A4  Calculation of C overflow

We assume that microbial biomass is strictly homeostatic, i.e. the C/N ratio of biomass $\left(\frac{C}{N}\right)_B$ is constant. This implies that the uptake of N from the soil solution must match the actual demand as defined by means of the C dynamics. It is calculated in

each biomass CAM cell as follows:

$$\left(\frac{N}{C}\right)_B \partial_t C_B = \left(\frac{N}{C}\right)_B (f_U - R_G - R_M - f_{BD}) = N_B\left((1-k_G)\nu_{C_S}\frac{C_S}{C_S + k_{C_S}} - k_M - k_{BD}\right) \tag{A1}$$

Depending on whether this amount of required N is available in the soil liquid at the current time step $t_k$, the following cases are distinguished at the discrete level and the overflow rate $R_O$ is defined:

$$R_O = \begin{cases} 0 & \text{if } \left(\frac{N}{C}\right)_B \frac{C_B^{t_k} - C_B^{t_{k+1}}}{\Delta t} < \frac{N_S^{t_k} - N_S^{t_{k+1}}}{\Delta t} & \text{with } N_S^{t_{k+t}} = 0, \\ \frac{C_B^{t_k} - C_B^{t_{k+1}}}{\Delta t} - \left(\frac{C}{N}\right)_B \frac{N_S^{t_k}}{\Delta t} & \text{if } \left(\frac{N}{C}\right)_B \frac{C_B^{t_k} - C_B^{t_{k+1}}}{\Delta t} \geq \frac{N_S^{t_k} - N_S^{t_{k+1}}}{\Delta t} & \text{with } N_S^{t_{k+t}} = 0. \end{cases} \tag{A2}$$

In the latter case the nitrogen pool $N_S$ in a cell is fully consumed in the subsequent timestep $t_{k+1}$. The overflow rate exactly balances the mismatch of demanded and available N, i.e. it accounts for a potential N limitation.

### A5 Variation of POM C:N = 10

Natural litter spans a range of C/N ratios. In order to limit the number of scenarios we did not vary the C:N ratio of POM in our study, but only the C:N ratios of exudates. The C:N ratio of 100 for POM is reasonable for fresh conifer or wood litter, but also for root litter. Poeplau et al. (2023) reported, e.g., C:N ratios from 50 - 124 there. We conducted an additional simulation using a lower C/N ratio for POM (C/N = 10) (see Fig. A1). As expected the results indicate that N limitation did not occur under these conditions; therefore, a higher amount of microbial necromass was formed, while the overall biomass size remained unchanged. This lines up consistently with the findings in the main manuscript.

*Author contributions.* MR designed and carried out the formal analysis, developed the methodology, implemented the software, prepared the visualizations, and drafted the original version of the manuscript. NR and AP contributed to the formal analysis, designed the methodological framework, secured funding, supervised the project, and drafted the original manuscript. EL, AS, and HM contributed to the conceptual design of the study, provided expertise and served as key contacts for questions related to the soil system, and critically revised and edited the manuscript. All authors have read and approved the final version of the manuscript.

*Competing interests.* The authors declare that they have no conflict of interest.

*Acknowledgements.* We gratefully acknowledge the financial support of the Deutsche Forschungsgemeinschaft (DFG) within the framework of the research unit 2179 "MAD Soil – Microaggregates: Formation and turnover of the structural building blocks of soils," project 276972051, and the priority programme 2089 (Rhizosphere spatiotemporal organization-a key to rhizosphere functions) under the project numbers 403660839 and 251268514. We also thank Ulla Rosskopf and Stephan Peth (Institute of Earth System Sciences, University of Hannover, Germany) for providing data on the particle size distributions. We thank the reviewers for their valuable comments.

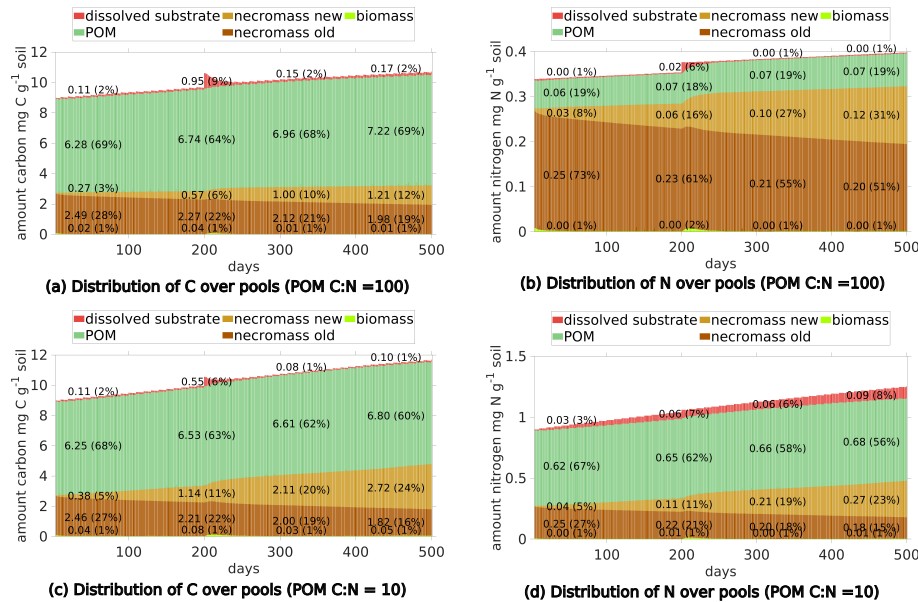

**Figure A1.** Comparison of C (left) and N (right) pools of two scenarios: top row: reference setting. bottom row: reference setting but with C/N ratio of POM = 10. In the bottom scenario the nitrogen limitation vanishes.

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
