# Peer review of "Coupled C and N turnover in a dynamic pore scale model reveal the impact of exudate quality on microbial necromass formation"

_EGUsphere, 2025_

## Author Comment (AC2)

**Referees' comments are in italics and black, while our responses are in blue.**

**Anonymous Referee #1, 30 Oct 2025**

We thank the reviewer for the valuable and conscientious comments to our manuscript. Please see the replies to the comments in detail below - we also prepare and submit a revised version of the manuscript.

Language. Section structure should be streamlined and harmonized, as now paragraphs are sometimes indented and sometimes not, several paragraphs contain a single sentence, and use of spaces between paragraphs is inconsistent. This makes it difficult to follow the narrative within each section. Please check citation format, as now it does not always follow standard practice ("Author (year)" for in text citations and "(Author, year)" for citations in brackets), e.g., L38, 174, 185, 286. Several sentences are hard to understand because of missing commas. To give one example, please see L199-202, where I would suggest adding four commas as follows: "Initially, POM particles with a concentration of 0.48 gCcm-3 and a C/N ratio of 100, and necromass with a concentration of 0.32 gCcm-3 and C/N ratio of 10, amounting to a volume fraction of 5\% of the solid area, were randomly added to the pore space, corresponding to 60 POM and 60 necromass particles, each with a size of 6-10 \$\mu\$m in diameter." In many instances word choice is not appropriate (though I am not a native speaker, so my impressions might be wrong), e.g., "Respired" instead of "Respirated" in Figure 1, L133 "outflows" instead of "sinks", L159 "Monod" instead of "Michaelis-Menten" (kinetics don't involve enzymes in this model), in some figure "amount" is actually a "content" (mass/mass). In other instances, the meaning of the chosen term is unclear, e.g., L44 "microbial C/N efficiency", L71 "experimental limitation", L401 "optimal" (what is optimized?), L558 "agitation", L570 "exclusive" (why are some pores exclusive?). These are just some examples. A thorough proof-reading is necessary, perhaps with help from a native speaker.

We thoroughly went through the manuscript and improved language as well as the arrangement of paragraphs and citations.

Methods structure. L74-93 present the model structure, so should be moved to the Methods. L260-270 do not present any result, but rather explain how model data is analyzed, so they belong to the Methods.

As suggested by the reviewer, we shifted the respective parts to the methods section.

Model implementation. It is not clear how the differential equations at the core of the model are solved in the hierarchical structure shown in Figure 4. Runge-Kutta method is used to solve the mass balance equations through time within a day, and then spatial fluxes are added, if I understand correctly. But if that is the case, is the solution converging numerically without successive iterations to feed back spatial flux information into the mass balance equations? Is the one-hour time step sufficiently short to ensure stability with an explicit method?

We use a non-iterative splitting approach (Line 585 in the submitted manuscript), where the solution of the diffusion equation is not iteratively feedbacked into the mass balance equation. Specifically, Equations (1), (2), and (5–8) are solved explicitly for each voxel using

a Runge–Kutta scheme. The resulting concentrations are then used as initial conditions to implicitly solve the diffusion equation. This is repeated 24 times within one day. To demonstrate that the chosen time step is adequate, we show the results of three additional simulations with doubled temporal resolution (48 equidistant steps per day) here. There are minor differences in absolute numbers but the same repartition of carbon among the different compartments. Note that due to the nature of the cellular automaton including some random movement of particles in the case of equal attractivity of several positions, there will always be (very small) differences in two simulations - even under the same conditions.

Figure: Compare C pool distribution of reference setting with 24 time steps per day (top left image) with different 3 simulations of the reference setting with 48 time steps per day (top right, and bottom row images). No significant difference can be observed.

Moreover, it is not clear how Eq. A2 is implemented. The inequalities compare partial derivatives on the left-hand side and contents on the right-hand side, which is not physically meaningful (units are different).

We define the concentration of nitrogen in soil solution within a voxel as  $N_s$ , which represents the maximum amount available for microbial assimilation during a given time step/moment  $t_k$ . In the subsequent the following timestep  $t_{k+1}$ , this nitrogen pool could be fully consumed (if the corresponding carbon pool demands this), such that  $N_s^{t_{k+1}}=0$ . As the decision rule is implemented discretely, we reformulated the equation accordingly:  $\left(\frac{N}{C}\right)_B \frac{C_B^{t_k} - C_B^{t_{k+1}}}{\Delta t} \geq \frac{N_s^{t_k} - N_s^{t_{k+1}}}{\Delta t} \quad \text{with } N_S^{t_{k+t}}=0$

Model setup. Some important assumptions make the simulations hard to generalize. First, the soil is assumed saturated for the duration of the simulations, but in such conditions, over 500 days anoxic conditions would develop, leading to rather different processes and controlling factors for C and N dynamics. While perhaps more difficult computationally, considering partly saturated conditions would make simulations more realistic (then one could argue that oxygen is not a limiting factor), and also allow comparing the effects of particle arrangement and pore water content on C and N dynamics. Which one dominates?

In our setup, we consider a domain of 500x500x1 micrometers which corresponds to a system of microaggregates, primary particles and micropores in a range of 2-80 microns. This does not necessarily mean that the whole surrounding soil needs to be saturated as larger pores beyond our scale would drain first. In our study we want to focus on the interplay of C and N with the microbial community, not limited by oxygen or water. The model framework could be extended in future work to study, e.g., the changes under anoxic conditions.

Second, POM is defined as particles with a size between 6 and 10 microns, while it is operationally defined as particles larger than about 50 microns. I agree that for this kind of numerical experiments, smaller POM particles are appropriate, but I would mention that they are smaller than according to the usual definitions.

We fully agree that, in the most usual definitions, POM of fresh litter typically refers to particles larger than approximately 50  $\mu$ m. In our simulations, however, we used smaller POM particles (6–10  $\mu$ m) to represent the fine particulate organic matter that is already partly decomposed and/or fragmented from larger POM particles as we are considering small scales. This choice allows us to capture microscale interactions within micro-aggregates between organic matter, microbes, and soil particles.

Figure: source is Figure 2 in [1]. POM can be also defined as particles smaller than 50 µm

More important, the C:N of POM is set to 100 g C/g N, which is reasonable for fresh conifer litter (or wood fragments), but much higher than most other plant residues. As a result, the simulated soil is almost always N limited, except at the time of root exudation, which provides inputs with lower C:N. I wonder if it should be the opposite—low organic matter C:N and high root exudate C:N?

We conducted an additional simulation using a lower C/N ratio for POM (C/N = 10). The results indicate that nitrogen limitation did not occur under these conditions; however, a higher amount of microbial necromass was formed, while the overall biomass size remained unchanged. This lines up consistently with the findings in the manuscript. In order to limit the number of scenarios, we did not vary the C:N ratio of POM there.

Figure: Compare C (left) and N (right) Pools of two scenarios. top row: reference setting. bottom row: reference setting but with C/N ratio of POM = 10. In the bottom scenario the nitrogen limitation vanishes.

Results presentation. Model simulations provide numerical values for fluxes and contents, but the values themselves are in general not particularly important and can be read from the figures. Now the Results section reports several numerical values that, in my view, distract from the main messages—what are the important trends and patterns we should look for in the figures? My suggestion is to streamline the Results section so that the answer to the research question (in this case a general aim rather than a question) emerges more clearly.

We see your point and streamlined the Results section.

Analysis of results. One of the most interesting results is briefly described in L373-383. There the interaction between spatial structure and organic matter stoichiometry emerges. I would expand this analysis beyond a qualitative statement based on visual inspection of a figure, making it quantitative. What can we say in general about these interactions across model runs within a scenario, or across scenarios?

Additionally to L373-383, in line 347-356 spatial results depicted in Figure 7 are discussed. There we describe the relation between the geometry of the pore space, the presence of microbial biomass and the availability of C using a normalized scale. Is there any other approach you would suggest to make the analysis more quantitative?

**Specific comments**

Thank you very much for your detailed comments. We revised the manuscript taking these comments into account.

**References**

[1] Jocelyn M Lavallee, Jennifer L Soong, and M Francesca Cotrufo. "Conceptualizing soil organic matter into particulate and mineral-associated forms to address global change in the 21st century". In: Global Change Biology 26.1 (2020), pp. 261–273.